# 3D Multi-Parameter Geological Modeling and Knowledge Findings for Mo Oxide Orebodies in the Shangfanggou Porphyry–Skarn Mo (–Fe) Deposit, Henan Province, China

**Zhifei Liu [1], Ling Zuo [1], Senmin Xu [2], Yaqing He [3], Chunyi Wang [3], Luofeng Wang [2], Tao Yang [2], Gongwen Wang [1,4,5,\*] , Linggao Zeng [6], Nini Mou [1] and Wangdong Yang [1]**

1   School of Earth Sciences and Resources, China University of Geosciences (Beijing), Beijing 100083, China; zfliu@cugb.edu.cn (Z.L.); 3001200120@cugb.edu.cn (L.Z.); nini@cugb.edu.cn (N.M.); yangwdcugb@163.com (W.Y.)
2   Luoyang Fuchuan Mining Co., Ltd., Luoyang 471500, China; xusm@cn.cmoc.com (S.X.); wangluofeng1@126.com (L.W.); kinkye@163.com (T.Y.)
3   China Molybdenum Co., Ltd., Luoyang 471500, China; heyq@cn.cmoc.com (Y.H.); wangcy@cn.cmoc.com (C.W.)
4   MNR Key Laboratory for Exploration Theory & Technology of Critical Mineral Resources, China University of Geosciences, Beijing 100083, China
5   Beijing Key Laboratory of Land and Resources Information Research and Development, Beijing 100083, China
6   No. 9 Geological Party, Bureau of Geo-Exploration and Mineral Development of Xinjiang Province, Urumqi 830000, China; lgzeng100@163.com
*   Correspondence: gwwang@cugb.edu.cn

**Abstract:** The Shangfanggou Mo–Fe deposit is a typical and giant porphyry–skarn deposit located in the East Qinling–Dabie molybdenum (Mo) polymetallic metallogenic belt in the southern margin of the North China Block. In this paper, three-dimensional (3D) multi-parameter geological modeling and microanalysis are used to discuss the mineralization and oxidation transformation process of molybdenite during the supergene stage. Meanwhile, from macro to micro, the temporal–spatial–genetic correlation and exploration constraints are also established by 3D geological modeling of industrial Mo orebodies and Mo oxide orebodies. SEM-EDS and EPMA-aided analyses indicate the oxidation products of molybdenite are dominated by tungsten–powellite at the supergene stage. Thus, a series of oxidation processes from molybdenite to tungsten–powellite are obtained after the precipitation of molybdenite; eventually, a special genetic model of the Shangfanggou high oxidation rate Mo deposit is formed. Oxygen fugacity reduction and an acid environment play an important part in the precipitation of molybdenite: (1) During the oxidation process, molybdenite is first oxidized to a $MoO_2 \cdot SO_4$ complex ion and then reacts with a carbonate solution to precipitate powethite, in which W and Mo elements can be substituted by complete isomorphism, forming a unique secondary oxide orebody dominated by tungsten–powellite. (2) Under hydrothermal action, $Mo^{4+}$ can be oxidized to jordisite in the strong acid reduction environment at low temperature and room temperature during the hydrothermal mineralization stage. Ilsemannite is the oxidation product, which can be further oxidized to molybdite.

**Keywords:** oxidation of molybdenite; ore-forming process; 3D multi-parameter geological modeling; microanalysis; Shangfanggou

## 1. Introduction

Three-dimensional (3D) geological modeling is a prominent technology that can be used to calculate or extract key parameters from 3D information on ore deposits [1]. Applying the theory of a metallogenic system, combined with multi-parameter or multi-source (geological, geophysical, geochemical, and hyperspectral) and multi-method modeling and

analysis of a 3D ore-forming geologic body, the capabilities of quantitative and big data collation, mineral exploration, and mining are greatly improved [2,3]. It has been proved that 3D GIS or modeling packages (e.g., Micromine, GOCAD, and Surpac) are excellent means of data representation and interpretation [4].

3D geological modeling provides distinct advantages in assisting geologists in determining the geological background, mineralization, and mineral exploration in a comprehensive and effective manner [5–18].

Porphyry–skarn-type molybdenum (Mo) deposits are one of the strategic and economic resources in China. The East Qinling Mo belt (EQMB; Figure 1B) is referred to as one of the most significant Mo provinces in the world, containing reserves of about 6 Mt Mo [19,20]. The Shangfanggou Mo–Fe deposit is an important deposit in the East Qinling Mo polymetallic metallogenic belt. It is adjacent to the Nannihu–Sandaozhuang Mo–W polymetallic deposit in the northeast (Figure 1D), which together constitute the main body of the Nannihu Mo ore field in Luanchuan district, China. In 2000, the estimated Mo reserves of the three major deposits (Nannihu, Sandaozhuang, and Shangfanggou) were about $2.4 \times 10^6$ tons (with an average Mo grade of 0.109%) [21], while the Shangfanggou deposit contained 0.72 million tons of Mo and 59.91 million tons of Fe metal, with average grades of 0.135% Mo [22,23] and 30.14% Fe [24].

Although there have been many studies on 3D geological modeling of the Luanchuan ore district [25,26], they have been insufficient to construct 3D multi-parameter geological modeling of the study area. Moreover, the formation of the Shangfanggou porphyry–skarn deposit is mainly controlled by strong tectonic magmatic movement in the Mesozoic. The complex geological background, multiple geological factors, and multi-scale and multi-format data sources pose challenges to 3D geological modeling and deep prediction.

Currently, the industrial Mo ores used by the mine are mainly the primary molybdenite; the onefold Mo oxide ores nearly cannot be recycled and utilized because they are difficult to beneficiate, and the ore-forming process (supergene oxidation process) of this kind of ore type also lacks research. In recent years, global Mo ore output has tended to decline, and the situation of rapid economic and social development has led to a demand for Mo. Therefore, it has become an important and urgent topic to study the genesis and prospecting of Mo ores in the oxidation zone and to utilize Mo oxide ores comprehensively and efficiently. On the basis of previous studies, this paper systematically discusses the mineralization and oxidation process of Mo orebodies in the Shangfanggou porphyry–skarn Mo–Fe deposit. The temporal–spatial–genetic correlation and exploration constraints of magnetite and gangue minerals to Mo and Mo oxide orebodies are discovered, and the distribution characteristics of the oxidation zone are further investigated (Figure 2). These findings provide a reference for the genesis and exploration, as well as the recovery and utilization, of oxidized ores for mining and beneficiation of the mine.

## 2. Geological Setting

### 2.1. Regional Geology

The EQMB is part of the Central China orogenic belt which is bounded by the San-Bao Fault to the north and the Shang-Dan Fault to the south [27,28] (Figure 1B). As the "Mo capital of China", more than 30 polymetallic deposits and ore spots are densely distributed in the Luanchuan ore district. This is consistent with the horizontal zoning of geochemical anomaly elements in the district. Namely, they are regularly distributed from inside to outside around the porphyry pluton: the porphyry–skarn Mo–W deposits and skarn sulfur polymetallic deposits mainly developed in the central zone of geochemical anomalies, while the hydrothermal vein Pb–Zn–Ag deposits are mainly distributed in the middle zone of geochemical anomalies [29]. Of course, both the ore district scale and the deposit scale have favorable prospecting potential at depth.

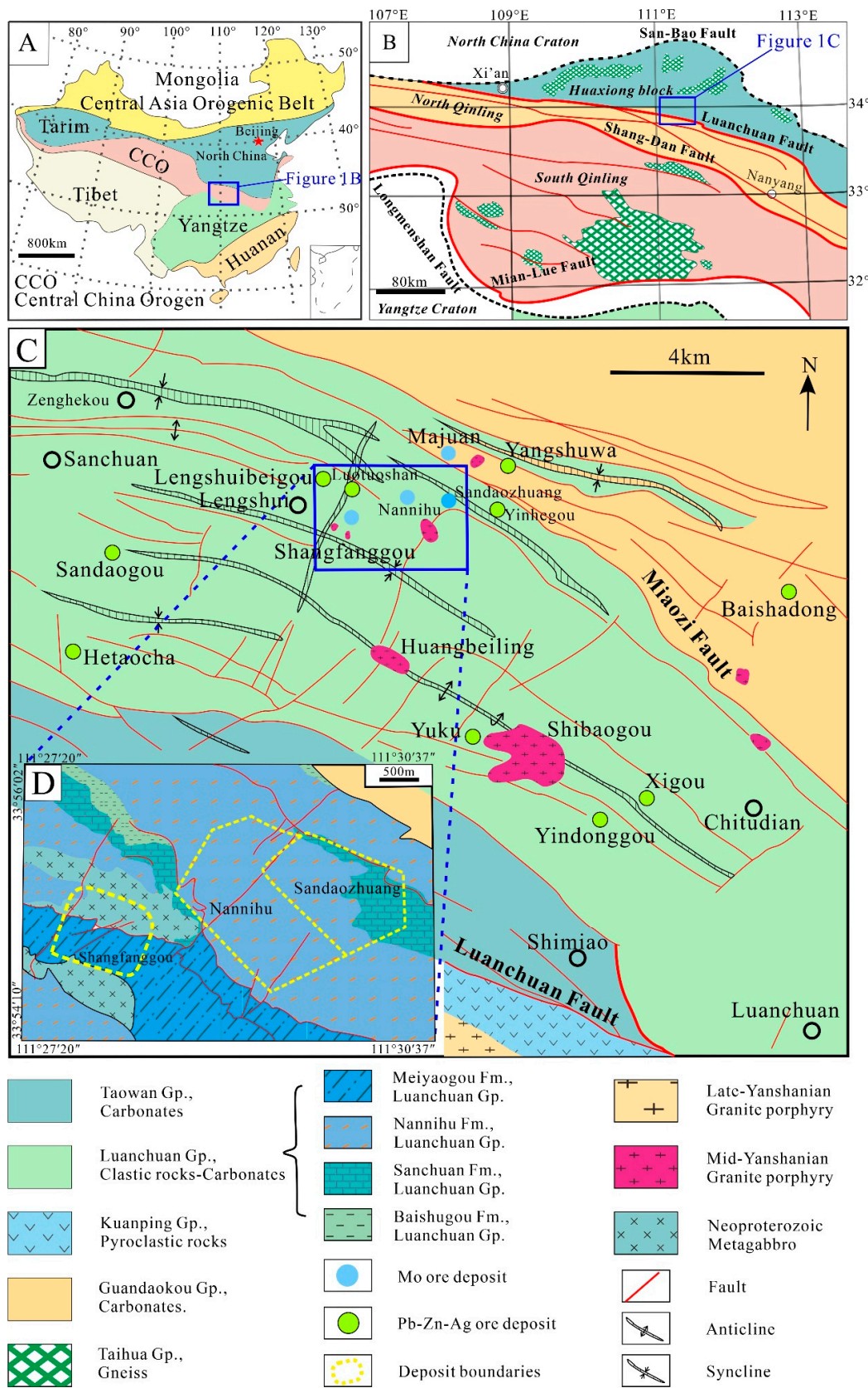

**Figure 1.** (**A**) Tectonic map of China, showing the location of the Qinling Orogen Belt; (**B**) tectonic subdivision of the Qinling Orogen Belt, showing the location of the Luanchuan ore district; (**C**) geological map of the Luanchuan ore district, showing the Shangfanggou Mo–Fe deposit (modified after [39]); (**D**) sketch of the Shangfanggou, Nannihu, and Sandaozhuang deposits (modified after [22,23]).

The Luanchuan ore district has a typical double-crust structure composed of basement and sedimentary cover. The basement is the Neoarchean Taihua Group deep metamorphic rock series, and the caprock series is mainly the "trident-rift" volcanic formation of the Middle Proterozoic Xiong'er Group [30], the Guandaokou group of the Middle Proterozoic, the passive continental clastic rock formation of the Luanchuan Group of the New Proterozoic, and the clastic rock–carbonates sedimentary formation of the slope facies of the Taowan Group, Lower Paleozoic [31]. The Nannihu Mo ore field, discovered in the 1960s and 1970s, is located in the north of the ore district, which is one of Mo's major places of production in China and even the world. The Nannihu Mo ore field, the Shangfanggou Mo–Fe deposit included, is mainly hosted in the Luanchuan Group.

Magmatism mainly concentrated in the Neoarchean, early Mesoproterozoic, and Mesozoic. Thereinto, the Mesozoic (Yanshanian) mineralization related to small porphyry was intense [32]. The main Mo mineralization is involved with the late Mesozoic granitic magmatism in the EQMB [33–36], forming giant porphyry–skarn Mo deposits, including the Nannihu, Shangfanggou mine, etc.

The main regional faults are the Luanchuan and Machaoying faults, which are distributed in a NWW-trending, and there are NE-trending secondary faults. These faults control the distribution of magmatic rocks and ore deposits, and giant deposits are often formed at the intersection of them [37,38]. That is, a series of NW-trending thrusts at a regional scale controlled the formation of major Mo deposits, which were formed by the regional Indosinian Qinling orogenic events. The secondary NW-trending folds and NE-trending faults and intrusive stock structures were formed by the thrusts during the Caledonian–Indosinian orogenic event. They are ore-bearing belts and ore-forming structures [2].

### 2.2. Deposit Geology

The Shangfanggou porphyry–skarn Mo–Fe deposit, located in the northwest of Luanchuan County, is world-renowned for its large scale, high grade, and suitability for open-pit mining. It is adjacent to the Sandaozhuang–Nannihu Mo–W polymetallic deposit in the northeast, constituting the main part of the Nannihu ore field in Luanchuan, a part of the world-famous district (Figure 1D). The deposit is located in the Sanchuan–Luanchuan fold belt in the southern margin of the North China Block, controlled by the Zenghekou–Shibaogou syncline. The genesis of the deposit is closely related to structure, granite porphyry pluton, wall rocks, strata (dolomite marble, skarn), and alteration, which are mainly distributed in NW- and NWW-trending (Figure 1C,D and Figure 2).

Metasediment of the upper Nannihu and the Meiyaogou Formation of the Luanchuan Group mainly crop out in the area, which belongs to the sedimentary environment of a shallow sea shelf-confined platform supratidal–subtidal zone. Under the influence of contact thermometamorphism and contact metasomatism caused by the intrusion of the Shangfanggou pluton, various types of hornfels and skarnization developed, respectively (for details of the lithology, see Section 3.1). Moreover, the alteration of wall rocks is composed of potassic alteration, silicification, and phlogopitization (Figures 2 and 3). With the intrusion of the pluton, the magnesian skarn formed from the dolomite marble of the middle Meiyaogou Formation (divided into three layers), which is the predominant ore-hosted position.

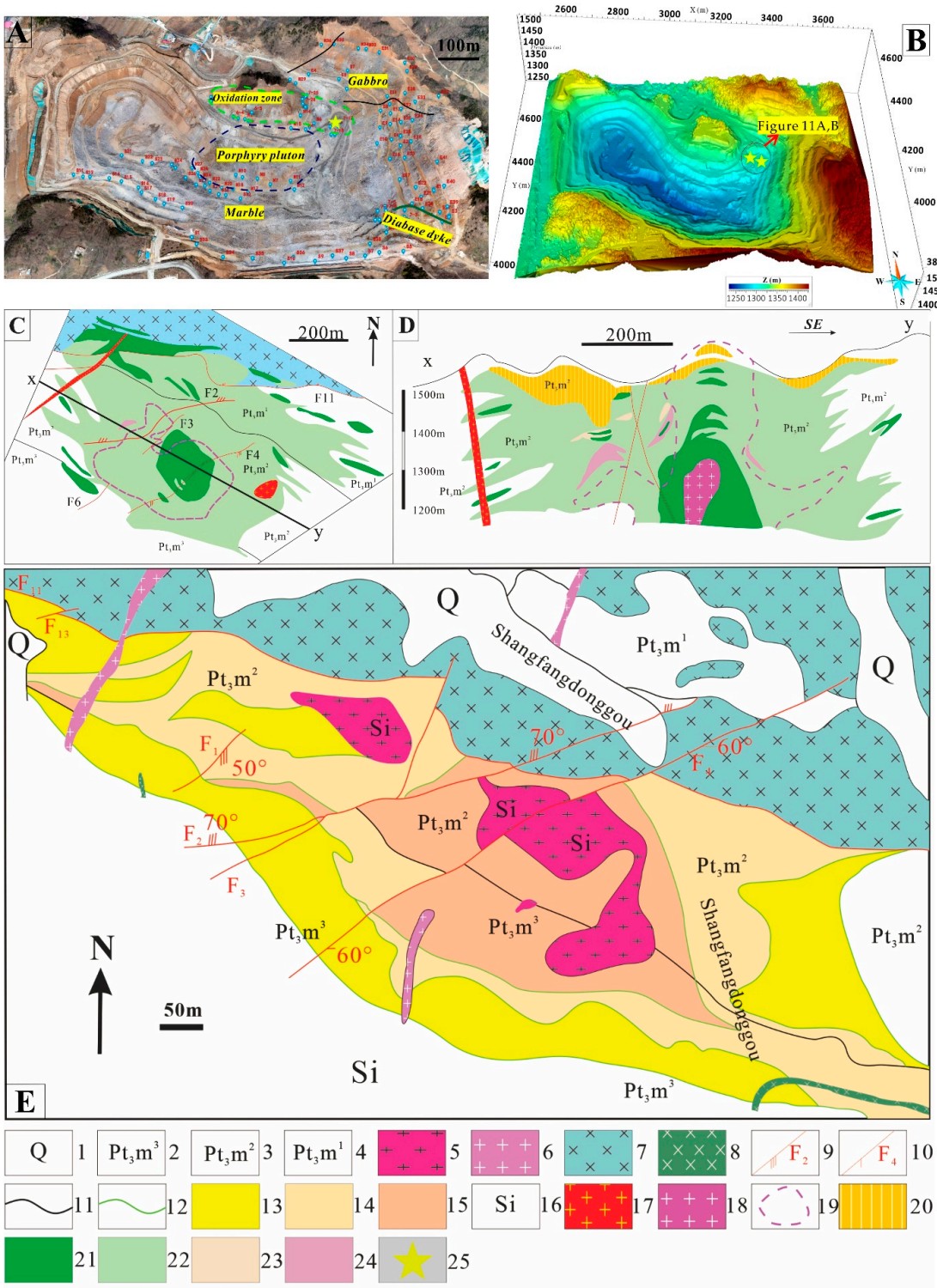

1. Quaternary; 2. Upper Meiyaogou Fm.; 3. Middle Meiyaogou Fm; 4. Lower Meiyaogou Fm.; 5. Granite p--orphyry; 6. Granite porphyry dike; 7. Gabbro; 8. Diabase; 9. Compression fault; 10. Transtension fault; 11. Geological boundary; 12. Alteration zone boundary; 13. Magnetite-tremolitization-diopsidization zone;14. Serpentinization zone; 15. Phlogopitization-actinolitization zone; 16. Strong silicification zone; 17. Post---ore porphyry; 18. Mineralized syenogranite porphyry; 19. Border of porphyritic biotite granite; 20. Oxidi--zed Mo orebody; 21. Low grade Mo orebody; 22. High grade Mo orebody; 23. Low grade Fe orebody; 24. High grade Fe orebody; 25. Sample location.

**Figure 2.** Geological sketch and location of Shangfanggou Mo–Fe deposit: (**A**) the open pit (red points, sample spots collected and analyzed in the fieldwork); (**B**) DEM of the mine; (**C,D**) geologic profile of the deposit, showing the Mo oxide orebodies and geology at 1132 m level and prospecting line NO. 5 (modified after [22,23,40–42]); (**E**) geological map of the deposit (after [43]).

Magmatism in the mining area is controlled by the fault structure, which mainly includes Caledonian metagabbro (in the south and north), syenite porphyry, and Yanshanian intermediate–acid granite porphyry (central part of the mining area). The metagabbro dikes yield K–Ar ages of ca. 743 Ma [44] and zircon SHRIMP U-Pb ages of ca. 830 Ma [45], which are associated with the rifting background at the southern margin of the North China Continent [46]. Therefore, the intrusive, alkali–feldspar granite porphyry in the middle Yanshanian, a time when the predominate mineralization was undergoing (Shangfanggou pluton, distributed in the center of the study area), is related to metallogenesis.

The deposit is controlled by the Shangfanggou syncline, which is located in the northern wing of the Shangfanggou syncline in the Sanchuan–Luanchuan County depression fold fault belt. NW–NWW- and NE–NNE-trending faults are developed, and the former controls the distribution characteristics of the plutons in the area, while the latter provides the conditions for migration and accumulation of Yanshanian ore-bearing hydrothermal fluid as a migration pathway. The NW–NWW trending dominates in magmatic rocks and various structures in the area, superimposing the later NNE–NE-trending faults. The NWW–NW-trending structure is mainly composed of the Zenghekou–Shibaogou syncline and a series of thrust nappe faults with NWW trending, in which the thrusts with southward thrusting and NE dip are developed. NNE–NE faults are intruded by the granitic porphyry dikes in both directions that control the formation of the Mo–Fe deposit and the intersection area of the two different directions is the occurrence site of the Shangfanggou orebody.

The Shangfanggou Mo–Fe polymetallic deposit occurs in the inner and outer contact zone of the alkali–feldspar granite porphyry. That is, the Mo ore bodies are hosted not only in the magnesium skarn but in the porphyry pluton and adjacent metagabbro and hornfels, which shows the controlling effect of the pluton on the deposit, while the Fe ore bodies occur in the former more often.

The ores can be divided into skarn, granite porphyry, hornfels and metagabbro types, according to the different host rocks (Figure 3A–G). The ore minerals consist of molybdenite, magnetite, and pyrite, followed by pyrrhotite, chalcopyrite, sphalerite, galena, etc. The gangue minerals are mainly quartz, diopside, garnet, tremolite, talc, serpentine, chlorite, feldspar, followed by sericite, carbonates, fluorite, etc. (Table 1; after [47]).

**Table 1.** Stages of mineralization and paragenesis for the Shangfanggou Mo-Fe deposit.

| Stages | Minerals |
| --- | --- |
| Protolith | Dolomite, K-feldspar, plagioclase, quartz, biotite, ilmenite, wollastonite |
| Late magmatism (silicification and potassic alteration) stage (1) | K-feldspar, quartz, biotite . . . |
| Early skarn stage (2) | Garnet, diopside, forsterite . . . |
| Late skarn stage (3) | Tremolite, magnetite, phlogopite, serpentine, calcite, actinolite, chlorite, talc, fluorite, serpentine . . . |
| Hydrothermal stage (4) | Quartz, pyrite, molybdenite, K-feldspar, chalcopyrite, galena, sphalerite, epidote, scheelite, fluorite, pyrrhotite . . . |
| Supergene stage (5) | Molybdite, ilsemannite, scheelite, limonite, tungsten–powellite . . . |

The gabbro in the northern part of the mine serves as part of the main orebody that is affected by mineralization. The Mo orebody occurs in (meta-)gabbro as well, accompanied by magnetite. In this kind of ore, pyroxene alteration is developed and widespread, which can be altered into amphibole and chlorite, giving rise to the partially argilled surface, and pyrite is mostly weathered into limonite (Figure 3G,H).

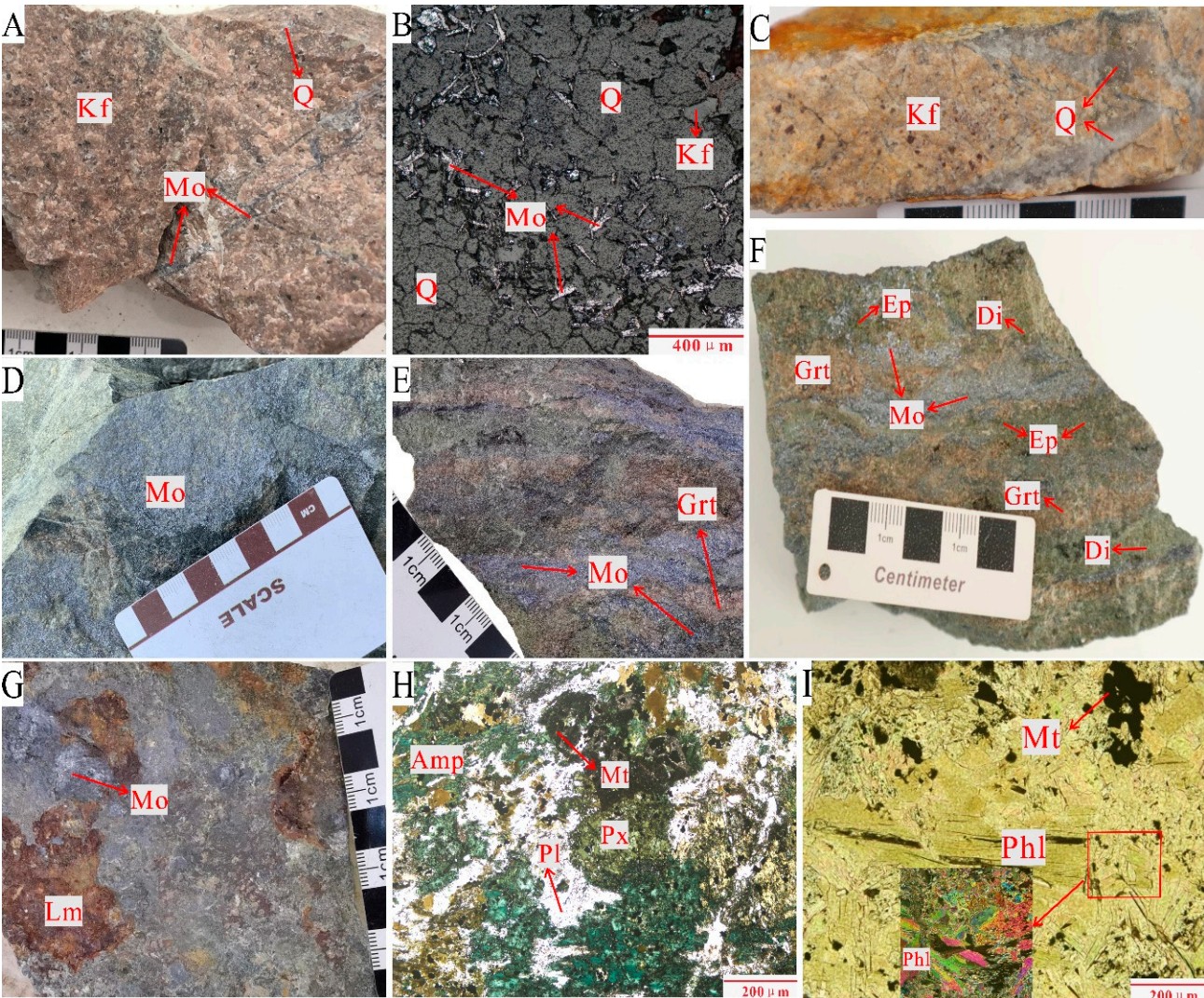

**Figure 3.** Photographs of alteration and mineralization characteristics in different ore types: (**A**) porphyry Mo ore and potassic alteration; (**B**) molybdenite with leaf-like texture unevenly embedded in gangue minerals such as quartz and K-feldspar or in veinlets in granite porphyry; (**C**) quartz veins (silicification) and potassic alteration in the granitic porphyry; (**D**) marble-type Mo ore; (**E**) skarn-type Mo ore with disseminated structure; (**F**) garnet diopside skarn-type Mo ore with epidotization; (**G**) metagabbro-type Mo ore with ferritization; (**H**) the alteration of pyroxene to amphibole in metagabbro and contiguous magnetite developed; (**I**) phlogopitization (colorful interference color of phlogopite).

The skarn-type Mo ores are distributed around the granite porphyry in the hanging wall (south) and on both sides of the east and west of the granite porphyry pluton, with a small amount of rock footwall, which is the dominant ore type of the deposit. Stellate and disseminated Mo mineralization can be observed, showing obvious metasomatism among this kind of ore. Molybdenite often occurs in the intergranular or fissure of skarn minerals such as diopside and garnet (Figure 3E,F).

The ore is dominated by a veinlet stockwork structure, followed by a disseminated structure. The veinlet structure is the most widespread and predominant structure in the area; the veinlet is not straight and has poor continuity. The ore texture is dominated by a granular texture, followed by a metasomatic relict texture (Figure 4).

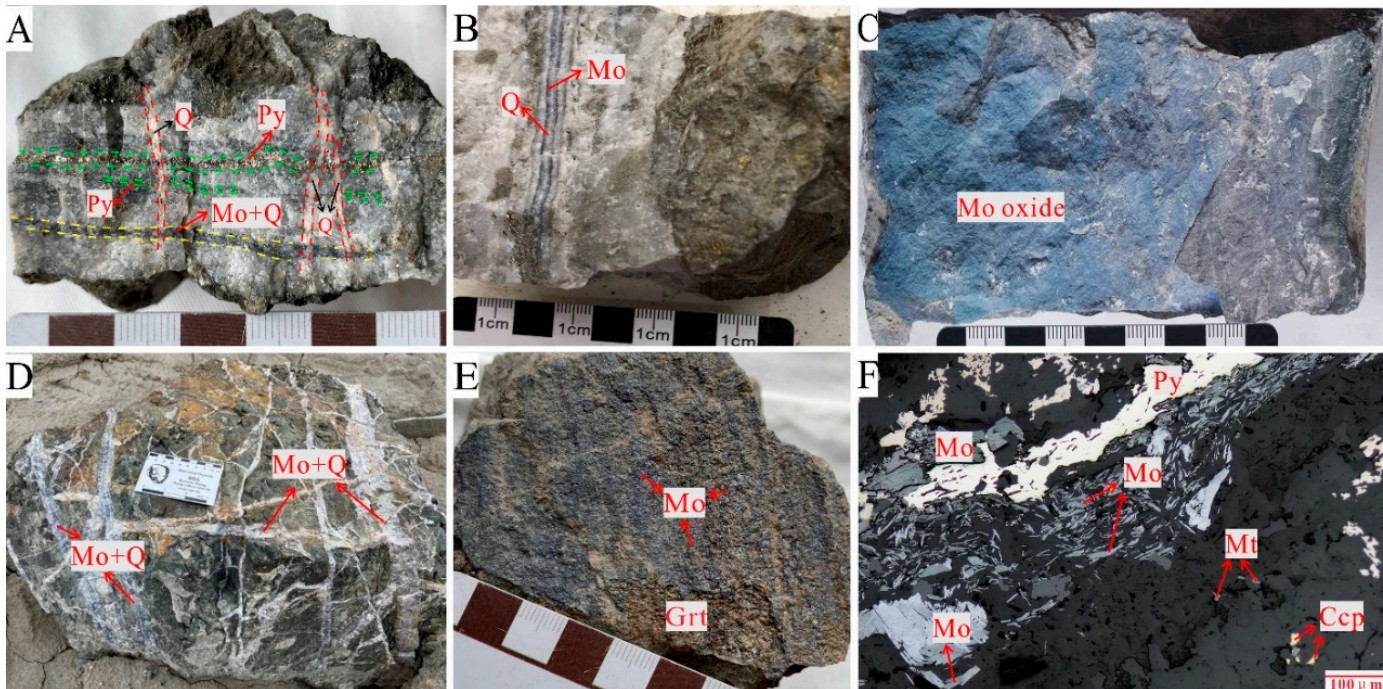

**Figure 4.** Photographs of ore petrography of the Shangfanggou Mo–Fe deposit: (**A**) quartz–molybdenite veinlet and pyrite veinlets crosscut by the late quartz veinlet; (**B**) banded structure; (**C**) membranous structure of the Mo oxide ore; (**D**) stockwork structure showing multistage mineralization; (**E**) disseminated Mo mineralization in the garnet-skarns; (**F**) the curved lepidosome molybdenite and paragenetic pyrite, magnetite, and chalcopyrite unevenly distributed between gangue minerals. Abbreviations: Q, quartz; Mo, molybdenite; Py, pyrite; Grt, garnet; Mt, magnetite; Ccp, chalcopyrite.

## 3. 3D Geological Modeling

Construct 3D geological model at the deposit scale using a variety of datasets such as strata, lithology, regional and deposit structures, geological section, boreholes, ore grade, and so on. 3D geological modeling usually consists of the following three steps: (i) geoscience data extraction and processing, (ii) 3D geological modeling, and (iii) interpretation and verification [1].

### 3.1. 3D Lithology Modeling

In GOCAD software, a 3D model of the lithology is established based on the datasets of geological section and stratum histogram. In Figure 5, the lithology of the three strata of the Luanchuan Group's Meiyaogou Formation can be visually recognized: the lower member is quartzite intercalated with mica schist and marble, etc., the middle part is thick dolomite marble, and the upper member is mainly dolomite marble. Granite porphyry pluton is distributed in the central part of the mining area, which is irregularly divided by faults and wall rocks on the surface and has a distinct contact boundary with the wall rocks.

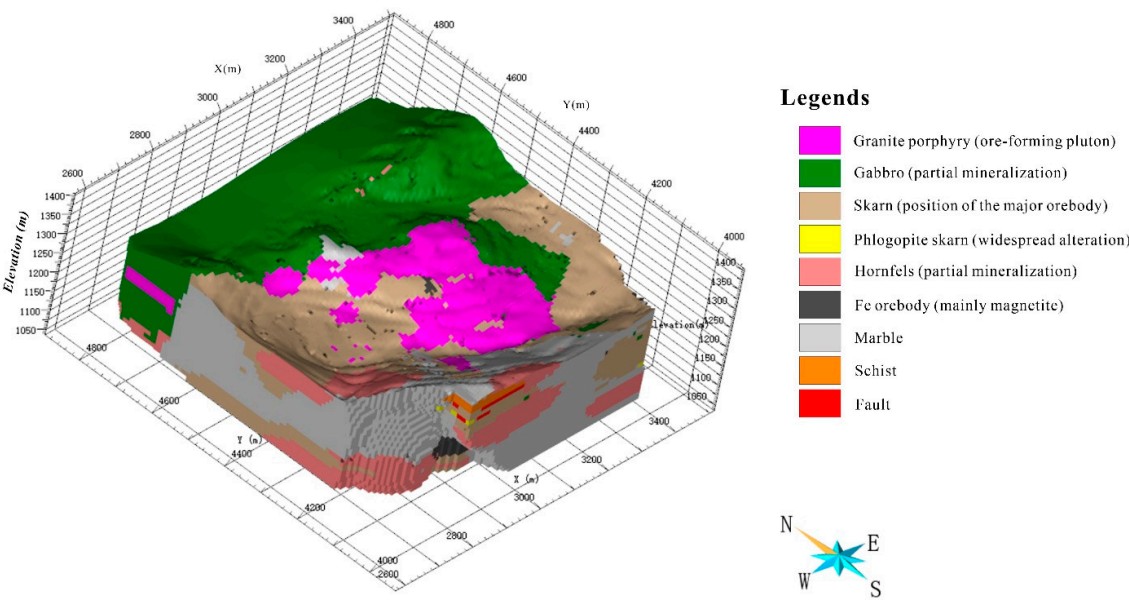

**Figure 5.** 3D lithology model of the Shangfanggou deposit.

### 3.2. 3D Structural Model

A structural model is established mainly based on the profile of prospecting lines and adjusted and verified by the horizontal geologic map. The software is mainly divided into the following four parts: fault, outline, contact, and surface. It is important to note that the definition of fault properties is critical.

The NWW–NW-trending faults dominating the study area and a series of NW-trending thrust structures control the formation of the main Mo orebodies (Figure 6). They are ore-bearing belts and ore-forming structures [2]. Moreover, NNE–NE trending also controls the formation of the deposit.

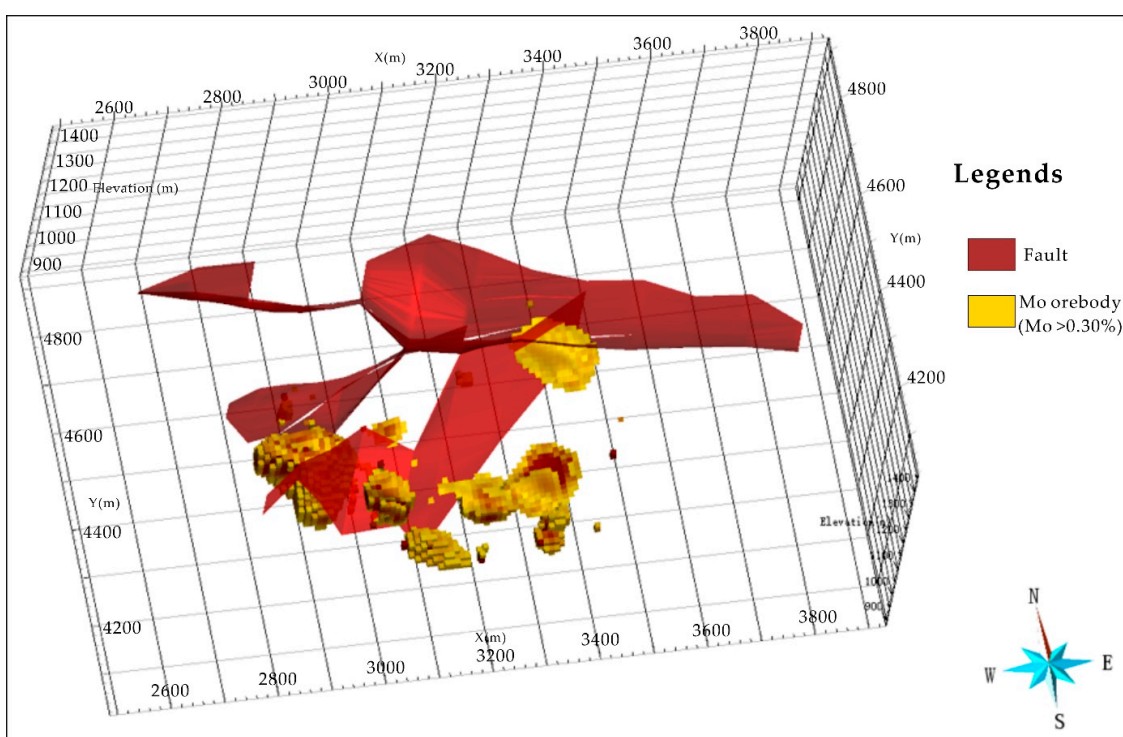

**Figure 6.** Spatial relationship between rich orebody and main faults (superposition of main faults with ore shoot).

### 3.3. 3D Orebodies and Grade Modeling

3D orebody geological modeling involves the delineation of the alteration zone, the identification of centers and boundaries of the orebody, and the mapping of fractures and faults. 3D orebody grade modeling is based on the borehole dataset through the application of indicator kriging (IK), a geostatistical interpolation tool from the GOCAD software [48]. In the case of the datasets with a favorable structure and a large amount of data, kriging interpolation with variogram function is more advantageous. The variation function has a good effect on analyzing the continuity of spatial data. Indeed, Mo orebodies are controlled by the diverse structures in the deposit, and the 3D orebody grade model is beneficial for inferring the geological events associated with mineralization. In addition, 3D orebody grade modeling by IK is helpful in identifying and extracting the different orebodies in one deposit [49,50].

The primary orebody is distributed in the form of an 'inverted cup,' with Mo oxide orebodies on the shallow surface. The orebody's border is uneven, and there are several branchings on the east and west sides. The thickness of the orebody varies on both sides, with the southwest being thinner and the northeast being thicker. The Mo orebodies are basically distributed above 600 m elevation but are mainly concentrated above 900 m, and the Mo oxide orebodies are typically found over 1300 m.

The skarn-type Fe orebodies are distributed in lenticular, irregular cystic, and saddle forms, with the shape determined by the occurrence of the pluton's contact zone. Analyzing both Figure 5 and Figure 8, the conclusion that can be drawn is that the depression and the roof of the pluton are favorable places for the enrichment of Fe mineralization. In the mine, the Fe orebodies are mostly found above 1000 m, with a downward trend, which is similar to the distribution characteristics of the Mo orebodies (Figures 7 and 8). Therefore, through the superposition of Mo mineralization, Fe, as an associated element, constitutes the Mo–Fe orebodies together with the Mo orebodies, indicating the spatial correlation between the Fe orebodies and the Mo orebodies and further speculating that the two are also related in genesis. The relative grade information of the two is shown in Figures 9 and 10.

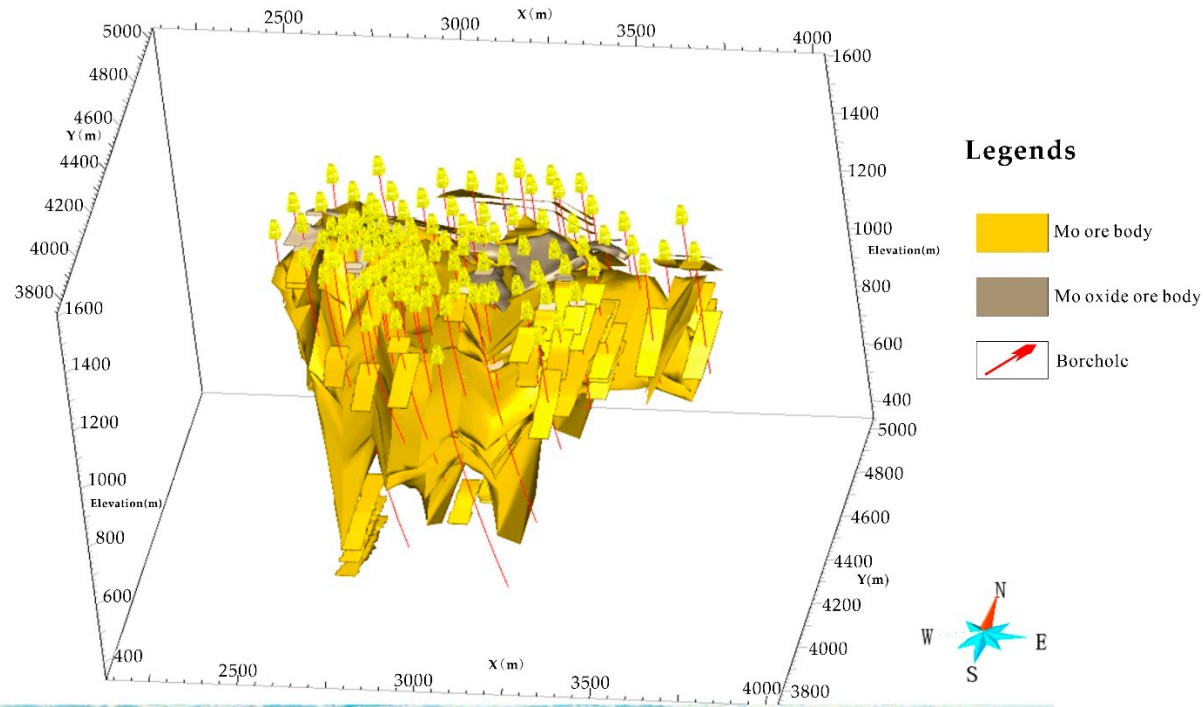

**Figure 7.** 3D model of Mo orebodies and Mo oxide orebodies.

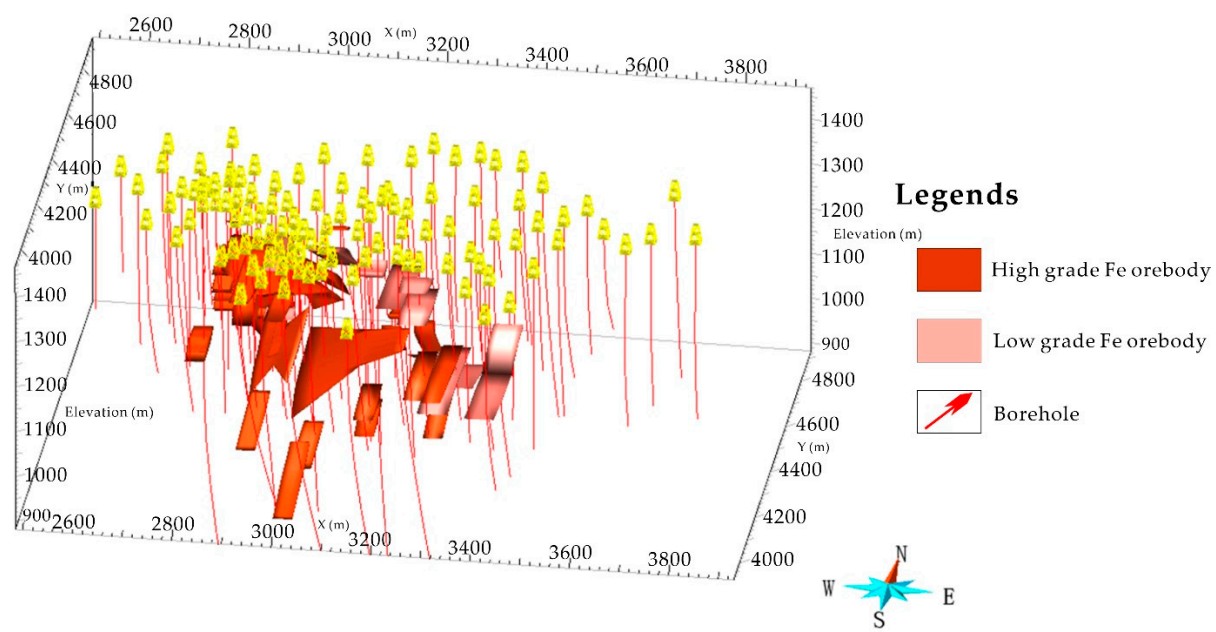

**Figure 8.** 3D model of Fe orebodies.

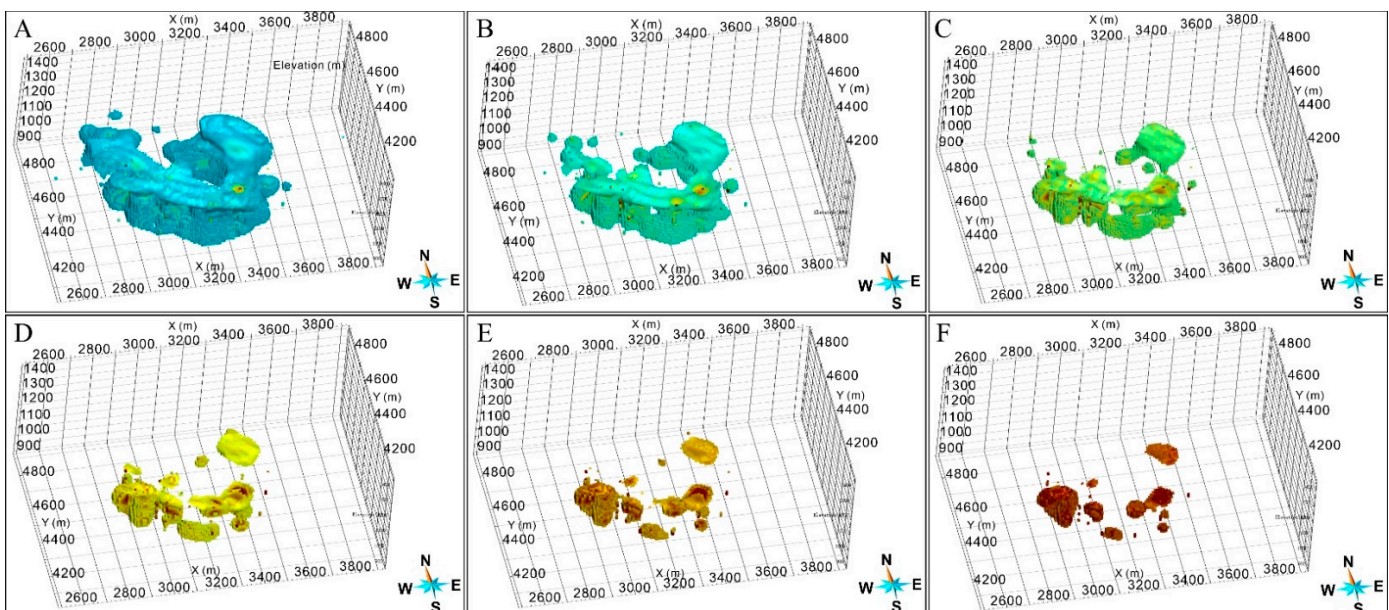

**Figure 9.** Distribution of Mo orebodies with different grades: (**A**) the grade > 0.10%; (**B**) the grade > 0.15%; (**C**) the grade > 0.20%; (**D**) the grade > 0.25%; (**E**) the grade > 0.30%; (**F**) the grade > 0.35%.

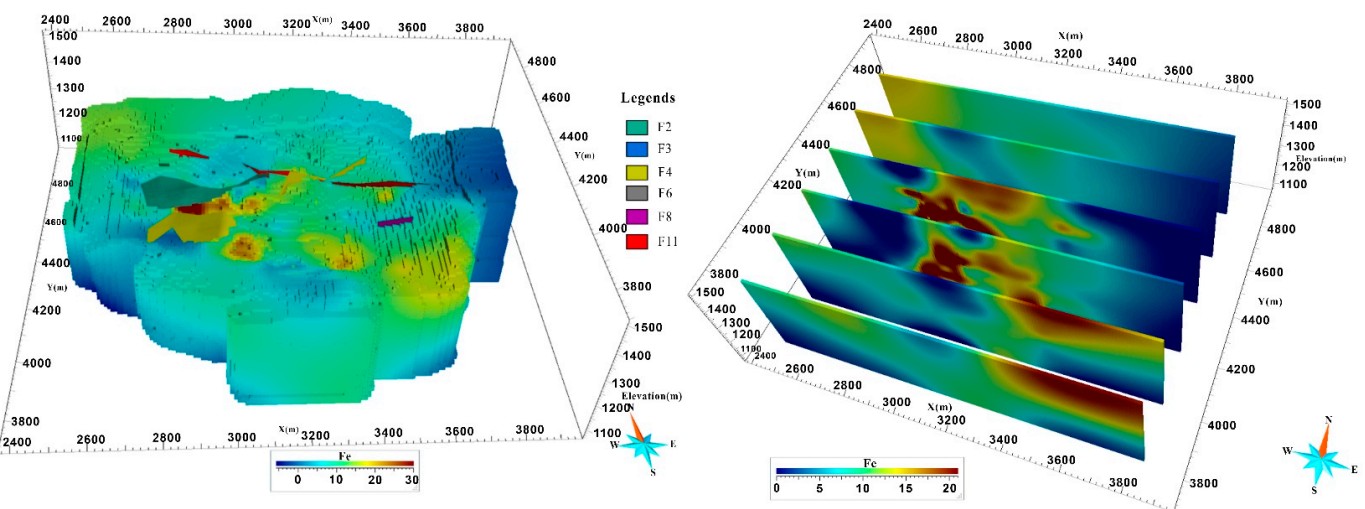

**Figure 10.** 3D grade model of Fe orebodies.

Figures [5], [7] and [9] illustrate that there is no ore in the core of the pluton. Although the orebodies are found on both sides, those toward the southwest are thick, concentrated, and of high grade.

The average grade of total Fe (TFe) in the Fe orebodies in the area is normally 20–30%, which is mostly concentrated in the hanging wall of the pluton or the top wall rock (magnesian skarn) within the range of 100–150 m from the intrusion contact zone of the pluton (Figure [10]).

## 4. Sampling and Analysis Methods

### 4.1. Sampling

Mo oxide ores were collected from the oxide zone of the deposit for microscopic identification (Figure [2]), SEM (scanning electron microscope)-EDS (energy dispersive X-ray spectroscopy), and EPMA (electron microprobe analysis) composition analysis.

The determination of major elements and high-resolution surface scanning and backscattering images were carried out at the Central Laboratory of Genetic Minerals, China University of Geosciences, Beijing, using FE-SEM Tescan. In addition, EPMA was performed at the Beijing Institute of Geology of Nuclear Industry, Beijing, China.

### 4.2. SEM-EDS

SEM-EDS was utilized to conduct a semi-quantitative analysis of molybdenite and its oxides, as well as the gangue minerals associated with them, in order to preliminarily explore their composition, content, and distribution (e.g., Figure [11]D–I). FE-SEM Tescan: MIRA 3 XMU field emission scanning electron microscope equipped with OXFORD X-Max 20 mm$^2$ X-ray energy spectrometer; backscattering electron image resolution was 2.0 nm@30 KV with beam drift ≤ 0.2% h; and energy dispersive spectroscopy was equipped with an electric refrigeration detector and peak to back ratio was better than 20,000:1. Moreover, Mn-ka is superior to 127 eV with an energy resolution of 20,000 CPS. The data were analyzed and processed using an Inca system with Inca X-Stream and Inca Mics microanalysis processors. The conditions of point analysis were as follows: high voltage, 20 KV; emission current, 83 µA; electron beam intensity, 18.36; absorption current, 4.0 nA; spot size, 85 nm; acquisition time, 30 s; process time, 3; dead time, less than 30%; count rate, higher than 20 Kcps.

### 4.3. EPMA

Representative measuring points were selected for more accurate component analysis of the samples preliminarily analyzed by SEM-EDS to determine the oxidation products of molybdenite: instrument (electron microprobe analyzer) model, JXA-8100. The im-

plementation standard is GB/T 15074-2008 "General Principles of Quantitative Analysis Methods for Electron Probe", China. In addition, the analysis conditions are as follows: an acceleration voltage of 20 kV was used for the experiment while the beam current was $1 \times 10^{-8}$ A and the exit angle was $40°$. Meanwhile, the spectral analysis method and ZAF correction method were adopted.

## 5. Discussion

### 5.1. Compositions

#### 5.1.1. EPMA

The composition data are listed in Table 2. The content of WO3 ranges from 11.21 to 32.40, with an average of 22.35 (unit: wt.%, and all of the following). The concentration of $MoO_3$ varies from 39.80 to 60.03, with an average of 49.64; it contains 24.18–27.08 CaO, with an average of 25.70 and a relatively uniform distribution; and the $SO_3$ content ranges from 1.18 to 1.93, with an average of 1.42.

**Table 2.** Representative analysis results (wt.%) of Mo oxide ores.

| Points | 1 | 2 | 3 | 4 | 5 | 6 | 7 | 8 | 9 | 10 | 11 | 12 | 13 | 14 | 15 |
|---|---|---|---|---|---|---|---|---|---|---|---|---|---|---|---|
| F | / | / | / | / | / | / | / | / | / | / | / | / | / | / | / |
| SiO₂ | / | / | / | / | / | / | / | / | / | / | / | / | / | / | / |
| WO₃ | 12.59 | 32.40 | 32.24 | 14.11 | 23.93 | 17.32 | 11.21 | 17.67 | 28.22 | 30.08 | 27.38 | 18.55 | 29.50 | 25.17 | 31.76 |
| SO₃ | 1.66 | 1.26 | 1.41 | 1.49 | 1.33 | 1.64 | 1.48 | 1.49 | 1.18 | 1.24 | 1.19 | 1.45 | 1.34 | 1.35 | 1.93 |
| Al₂O₃ | / | / | / | 0.02 | / | / | / | / | / | / | / | / | / | 0.02 | / |
| MgO | / | / | / | / | / | / | / | / | / | / | / | 0.13 | / | / | / |
| MoO₃ | 58.51 | 40.91 | 40.65 | 57.55 | 48.86 | 53.66 | 60.03 | 54.21 | 43.81 | 42.54 | 45.56 | 51.85 | 43.74 | 47.59 | 39.80 |
| CaO | 26.73 | 24.35 | 24.37 | 26.36 | 25.07 | 26.69 | 27.08 | 26.05 | 25.97 | 25.29 | 25.03 | 26.77 | 24.18 | 25.17 | 24.80 |
| SeO₂ | 0.10 | 0.63 | 0.66 | 0.14 | 0.51 | 0.26 | 0.13 | 0.31 | 0.44 | 0.77 | 0.64 | 0.34 | 0.54 | 0.60 | 0.63 |
| FeO | 0.07 | 0.10 | 0.05 | 0.03 | 0.12 | 0.03 | / | 0.06 | 0.11 | 0.07 | 0.07 | 0.14 | 0.20 | 0.09 | 0.06 |
| Cl | 0.16 | 0.08 | 0.11 | 0.19 | 0.15 | 0.14 | 0.13 | 0.16 | 0.14 | 0.10 | 0.12 | 0.13 | 0.14 | 0.12 | 0.07 |
| MnO | / | / | 0.04 | / | / | / | / | / | 0.04 | 0.07 | 0.03 | 0.04 | / | / | / |
| P₂O₅ | / | / | / | / | / | / | 0.06 | / | / | / | / | / | / | / | / |
| CuO | / | / | / | / | / | / | / | / | / | / | / | / | / | / | / |
| PbO | / | / | / | / | / | / | / | / | / | / | / | / | / | / | / |
| TeO₂ | / | / | / | / | / | / | / | / | / | / | / | / | / | / | / |
| Total | 99.82 | 99.73 | 99.53 | 99.89 | 99.97 | 99.74 | 100.12 | 99.95 | 99.91 | 100.16 | 100.02 | 99.40 | 99.64 | 100.11 | 99.05 |

/: Not detected.

Within the detection limit of the instrument, the mineral does not contain Cu, Pb, Zn, Si, F, Te, etc. Mn, Al, etc. were found on occasion. In terms of the contents, Mo is the predominant element in the oxide mineral granule, while Mo elements are mainly present in the tungsten–powellite.

The relative contents of Mo and Ca at the seventh site are the highest, which are 60.03% and 27.08%, respectively, while the content of W is the lowest at only 11.21%. A minor quantity of P is contained, and Fe, Mn, etc. are not included.

#### 5.1.2. SEM-EDS

The tungsten–powellite is finely granular and is irregularly dispersed in the oxide ores (Figure 12).

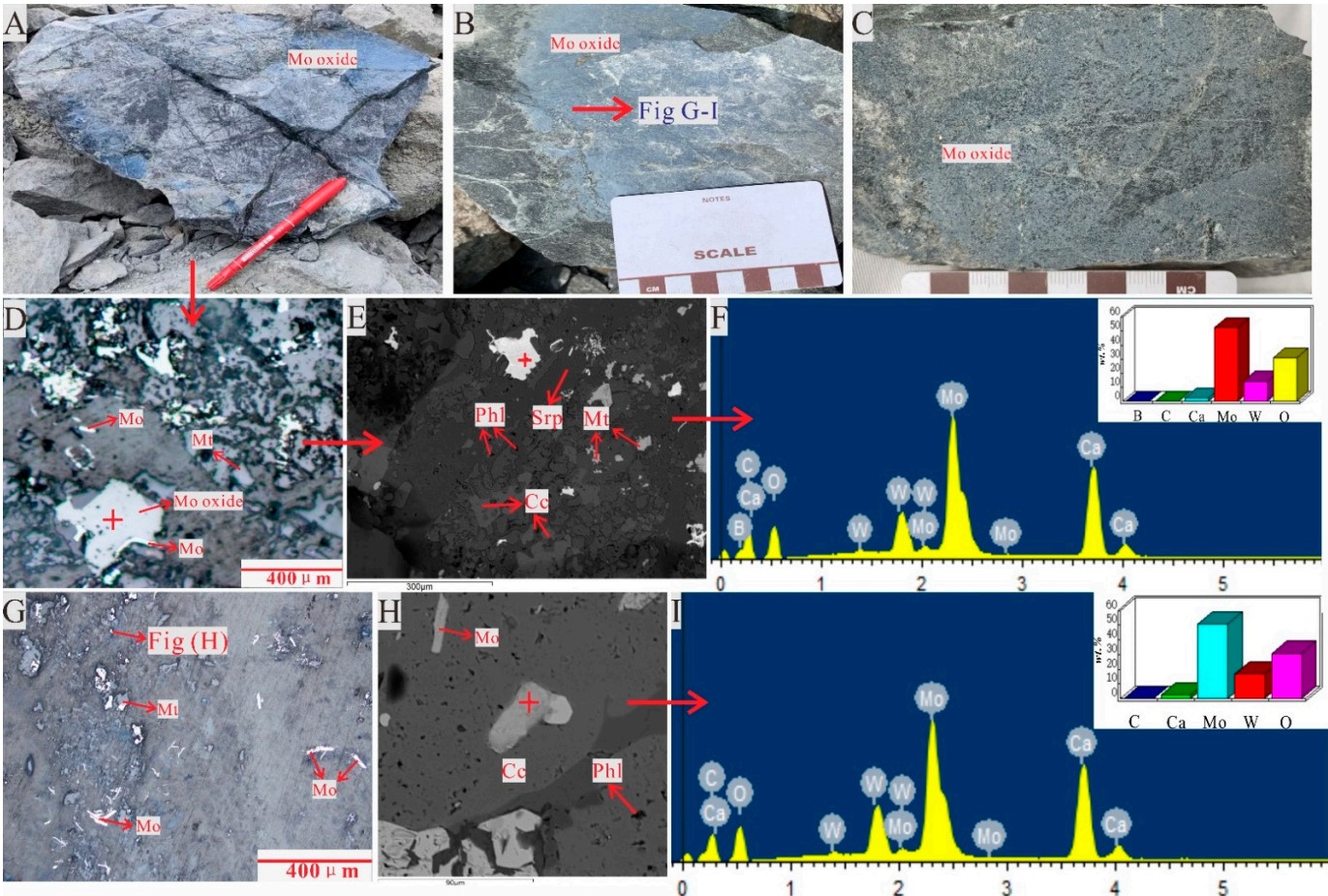

**Figure 11.** Photographs showing the characteristics of field and hand specimens of Mo oxide ores as well as SEM analysis results: (**A**–**C**) Mo oxide ores; (**D**,**G**) molybdenite and associated with magnetite; (**E**,**H**) the test points of EDS and BSE of the Mo oxide minerals and accompanying gangue minerals; (**F**,**I**) results of EDS. Abbreviations: Mo, molybdenite; Mt, magnetite; Cc, calcite; Phl, phlogopite; Srp, serpentine.

Mo oxide minerals are primarily found in the skarnized marble, which is usually associated with phlogopitization and serpentinization. The tungsten–powellite is found between the grains and veins of calcite or hydrothermal alteration minerals such as phlogopite and serpentine, indicating a later stage of tungsten–powellite development rather than the skarn stage or earlier.

In Figure 12, compared with Mo, the distribution of the W element is not uniform, and obvious Mo and W bands are visible, while for the element, Mo distribution is relatively uniform, and the Mo content of molybdenite in the (almost) unoxidized margin is higher than that in the middle of the mineral.

The element distribution of Ca, Fe, and O is uniform, while the content of Fe is significantly lower than that of Ca. In addition, S is mainly distributed in the margin molybdenite. All of these are consistent with the results in Table 2 and the characteristics of tungsten–powellite, indicating the incomplete oxidation or dynamic oxidation process of molybdenite. The contents of Mo, Ca, and W are higher than that of O, Mg, and Si, which further confirms the oxidation of molybdenite. The characteristics of the zonal concentration of W content also demonstrate the distribution of a W-bearing hydrothermal solution.

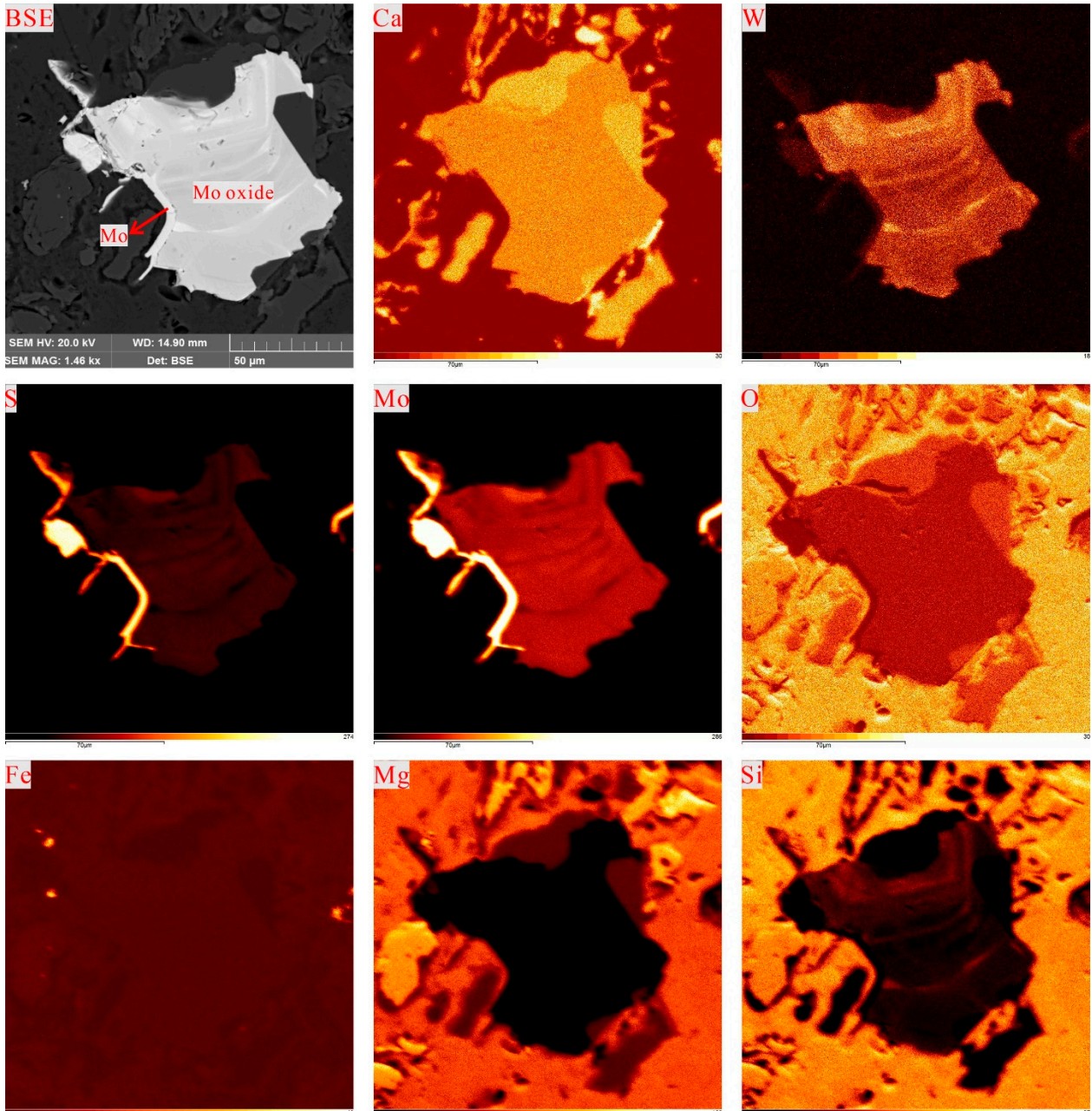

**Figure 12.** Mapping of the elements scanning results based on SEM. Abbreviation: Mo, molybdenite.

When molybdenite is not completely oxidized, it can be observed that the pseudomorph of molybdenite is retained in the granules and distributed along the edge of tungsten–powellite (Figures 11D and 12). Moreover, unoxidized molybdenite is sporadically present in the samples. Magnetite is disseminated and indicates a high oxygen fugacity environment.

## 5.2. Mineralization and Supergene Oxidation Process

Mineralization is intimately linked to temperature and oxygen fugacity ($f(O_2)$). Previous research has revealed that the ore-forming fluids of many of the East Qinling's magmatic–hydrothermal deposits are strong oxidizing fluids with high temperature, salinity, and oxygen fugacity [51–53].

The crystallization temperature and $f(O_2)$ of biotite in the Shangfanggou pluton are 750 °C–860 °C and −8.0–−6.5, respectively, indicating a relatively high temperature and

f(O$_2$) environment. The crystallization temperature and f(O$_2$) of biotite related to Mo mineralization are significantly greater than those of other ore-forming plutons. It is speculated that the pluton related to Mo mineralization also originates from a high temperature and f(O$_2$) environment [54].

The K–Ar, Rb–Sr, and zircon U–Pb ages of the Shangfanggou porphyry pluton are all between 134 and 158 Ma [33,55–58], similar to the Re–Os isotopic dating of molybdenite (metallogenic age, 143.8~145.8 Ma) [59], suggesting the deposit is of magmatic–hydrothermal genesis in Yanshanian. The Yanshanian magmatism provides hydrothermal and metallogenic material sources for Mo polymetallic mineralization in the study area [54]. Meanwhile, the magma source is mixed crust and mantle-derived (with I-type granite characteristics, rich in Mg and poor in Fe) [60].

The ore-forming fluid of the Shangfanggou Mo–Fe deposit is mostly magmatic water, with a tiny quantity of metamorphic water in the strata, according to H–O isotope research. The S isotope composition is rather consistent and is characterized by deep source sulfur, which might come from the pluton. The Pb isotopic composition of the Nannihu ore field granite porphyry is notably different from that of the Shangfanggou pluton, which may inherit the Pb isotopic composition features of the Shangfanggou pluton [61].

In general, during the mineralization stage, the fluid system evolved from oxidation to reduction, and the composition changed from complex to simple [51]. A simplified deposit exploration model is described in Figure 13.

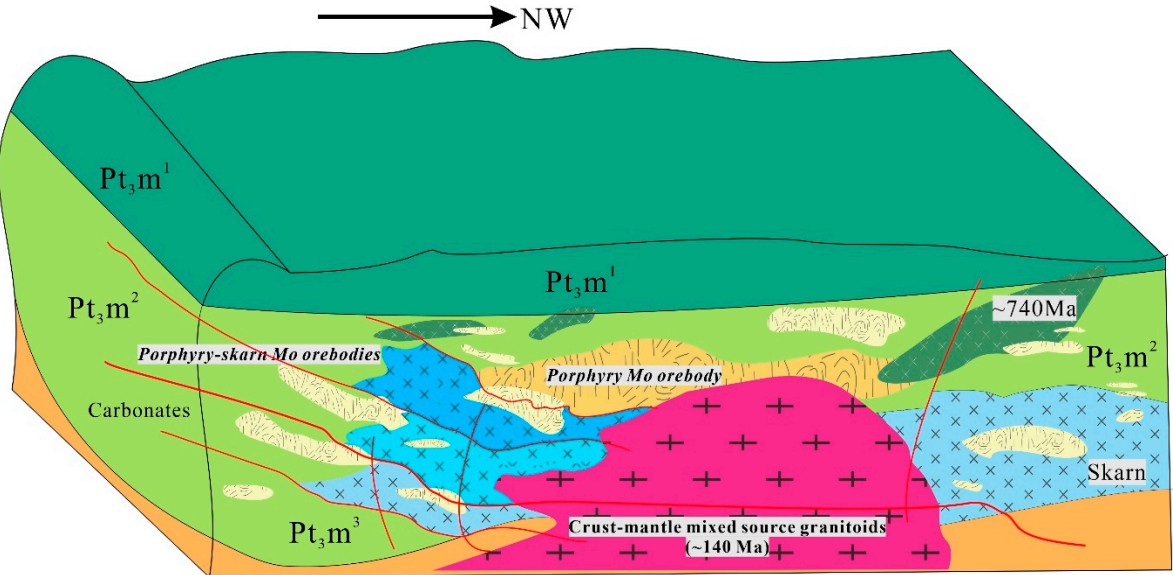

**Figure 13.** Sketch map of the Shangfanggou deposit exploration model.

To sum up, in such a metallogenic environment, a decrease in f(O$_2$) and an increase in sulfur fugacity play an important role in molybdenite precipitation. With local redox reaction or neutralization on the contact zone between the hydrothermal solution and carbonates, plenty of sulfides began to precipitate. At the high to medium temperature stage, a substantial quantity of molybdenites developed, resulting in high-grade orebodies dominated by molybdenite.

Under hydrothermal action, molybdenite precipitates under acidic conditions; that is, molybdenite is the most stable under acidic conditions. Under the condition of a supergene environment, molybdenite is oxidized, leached, and migrated with the medium in the state of a MoO$_2$·SO$_4$ complex ion, and then contacted with carbonate solution to precipitate powethite. Therefore, W and Mo elements can be completely replaced by complete isomorphism, forming unique secondary orebodies dominated by tungsten–powellite with a high oxidation rate in the Shangfanggou Mo–Fe deposit. The basic reaction process is as follows:

$$2MoS_2 + 9O_2 + 2H_2O = 2\,(MoO_2 \cdot SO_4) + 2H_2SO_4; \tag{1}$$

$$MoO_2 \cdot SO_4 + Ca\,(HCO_3)_2 = CaMoO_4 + H_2SO_4 + 2CO_2 \tag{2}$$

Moreover, powethite also metasomatized tungsten–powellite. The light and dark bands alternately formed by Mo-rich or W-rich bands in the mineral grains can be observed in SEM (Figure 12); the percentage of tungsten–powellite to total Mo changes depending on the oxidation degree of molybdenite.

While under hydrothermal action, $Mo^{4+}$ can be oxidized to jordisite in a strong acid reduction environment at low temperature and room temperature during the metallogenic epoch. Ilsemannite is the oxidation product, which can be further oxidized to molybdite.

## 6. Conclusions

Combined with the 3D model, (1) it can be confirmed that the surrounding rocks of the gently inclined upper contact zone and the top of the pluton are more conducive to the formation of high-grade orebodies, the form of Mo orebodies is mostly related to spatial morphology and the occurrence of small plutons, and there is no ore in the core of the pluton. (2) Molybdenite began to be generated after the precipitation of magnetite, resulting in the formation of Fe orebodies earlier than that of Mo, while the Fe orebodies normally formed Mo–Fe orebodies caused by the superimposed Mo mineralization. In general, magnetite-dominated Fe orebodies have a temporal, spatial, and genetic relevance and/or control effect on Mo and its oxide orebodies.

The Shangfanggou Mo–Fe deposit was formed in an environment of high temperature and high oxygen fugacity in Yanshanian, provided by a crust–mantle mixed source and magmatic–hydrothermal and ore-forming source. The ore-forming fluid is given priority to magmatic water, with the possible addition of a small quantity of metamorphic water from the strata. With the precipitation of molybdenite, it experiences the following two different oxidation processes with a decrease in temperature, oxygen fugacity, and acidity: (1) Supergene oxidation (Stage 5)—molybdenite-$MoO_2 \cdot SO_4$ complex ion–Mo oxide minerals aggregate dominated by powethite–tungsten–powellite. During the oxidation process, molybdenite is first oxidized to a $MoO_2 \cdot SO_4$ complex ion and then reacts with a carbonate solution to precipitate powethite, in which W and Mo elements can be substituted by complete isomorphism, forming unique secondary oxide orebodies dominated by tungsten–powellite. (2) Metallogenic epoch (Stage 4)—when the solution is strongly acidic, molybdenite is first transformed into jordisite, and the oxidation product is ilsemannite, which finally can be completely oxidized to molybdite.

**Author Contributions:** Conceptualization, G.W.; methodology, Z.L. and L.Z. (Ling Zuo); software, N.M. and W.Y.; validation, S.X., C.W., Y.H., T.Y. and L.W.; formal analysis, Z.L.; investigation, Z.L. and G.W.; project administration, L.Z. (Ling Zuo) and L.Z. (Linggao Zeng); data curation, Z.L. and G.W.; writing—original draft preparation, Z.L.; writing—review and editing, Z.L. and G.W. All authors have read and agreed to the published version of the manuscript.

**Funding:** This research was supported by Fuchuan Mining Co., Ltd., China Geological Survey (Grant No. DD20190570).

**Data Availability Statement:** All the data are presented in the article.

**Acknowledgments:** We are grateful to Yi Cao for his constructive comments and discussion. Many thanks are given to Xiaojiang Zhou from Luoyang Fuchuan Co., Ltd. for his kind assistance in the fieldwork. We also thank Leilei Huang and Yulin Chang for their help in data collection and processing, and for the laboratory assistance provided by Peipei Li of the Central Laboratory of Genetic Minerals, China University of Geosciences (Beijing) during the SEM. The valuable comments and suggestions from the four reviewers considerably improved the manuscript.

**Conflicts of Interest:** The authors declare no conflict of interest.

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
