# Peer review of "3D Multi-Parameter Geological Modeling and Knowledge Findings for Mo Oxide Orebodies in the Shangfanggou Porphyry–Skarn Mo (–Fe) Deposit, Henan Province, China"

_minerals, doi:10.3390/min12060769_

Round 1

Reviewer 1 Report

In this paper, three-dimensional multi-parameter geological modeling and microanalysis are used to discuss the mineralization and oxidation transformation process of molybdenite in the supergene stage. Meanwhile, from macro to micro, the temporal-spatial-genetic correlation and exploration constraints are also established by 3D geological modeling of industrial Mo orebodies and Mo oxide orebodies. The method and the presented results are basically convincing. All together it shows a clear and valuable contribution. However, the following minor issues need to be clarified for this manuscript to be publishable.

1. How to evaluate the reliability of the 3D models? Comparative analysis with known data or some quantitative statistical methods should be used to evaluate the reliability of the models.

2. Line 50: Three-dimensional (3D) -> 3D. It already appeared in the first paragraph.

3. Lines 87/130/140/185: Error! Reference source not found.

4. Lines 210/216/225/232: “3D”, 3 is missed.

Reviewer 2 Report

The manuscript reports the comprehensive study of Mo ore bodies in Henan Province. Detailed studies on the evolution and origin of ore bodies are presented followed by SEM-EDS and EPMA analysis and 3D geological modeling. The results reported represent a notable advance in the understanding of oxidation processes of molybdenite deposits.

The title and abstract are appropriate for the content of the text. Furthermore, the manuscript is well constructed and analyses are well performed. In abstract, the authors summarized the main research question and key findings. The description of study subject and results are detailed and correct. Data collected are interpreted accurately and the results support the conclusion.

This manuscript reads well, but needs the English editing. The figures are clear and readable and support the findings. The reference list needs to be completed. In my opinion, the manuscript is suitable for publication in Minerals with minor revision. All the remarks are listed in comments in PDF.

Reviewer 3 Report

Dear Authors,

I hope these minor suggestions would be useful for you.

I am not a native speaker. Thus, the text should be checked for possible syntax and grammatical errors.

In the text there are some statements these should be removed from the text (Error! reference source not found).

In some subtitles there are missing statements. D should be 3D.

For the benefit of the readers the modelling algorithm should be explained in a detailed manner.

In line 50 there is no need to write three-dimensional. Because you gave it in line 42.

Please improve the sentences between the lines 50 and 52.

In line 59 The must be the.

In figure 2b the axes values are too small to read. Please enlarge them. Figure 2a needs a distance scale.

In the Figures 5 and 6 the legends are off the pages.

In the figures 9 and 10 the axes values are too small to read. Please enlarge them.

Reviewer 4 Report

Without hesitation, it is a very interesting comprehensive investigation prepared at a high level.

However, there are some remarks.

1. A question: Does this deposit contains any other useful minerals of commercial value?

2. The Conclusions do not fully reflect the obtained results and should be improved.

3. I propose that in the Introduction, the following sources can be added as effective examples of 3D geological-geophysical modeling:

Alizadeh, A.A., Guliyev, I.S., Kadirov, F.A., and Eppelbaum, L.V., 2017. Geosciences in Azerbaijan. Volume II: Economic Minerals and Applied Geophysics. Springer, Heidelberg – N.Y., 340 p. 

Mónica Arias,  Pablo Nuñez,  Daniel Arias, Pablo Gumiel, Cesar Castañón, Jorge Fuertes-Blancoand Agustin Martin-Izard, 2021. 3D Geological Model of the Touro Cu Deposit, A World-Class Mafic-Siliciclastic VMS Deposit in the NW of the Iberian Peninsula. Minerals, 11 (1).
